# EGFR Exon 20 Insertion in Metastatic Non-Small-Cell Lung Cancer: Survival and Clinical Efficacy of EGFR Tyrosine-Kinase Inhibitor and Chemotherapy

**DOI:** 10.3390/cancers13205132

**Published:** 2021-10-13

**Authors:** Samy Chelabi, Xavier Mignard, Karen Leroy, Isabelle Monnet, Solenn Brosseau, Nathalie Theou-Anton, Marie-Ange Massiani, Sylvie Friard, Boris Duchemann, Elizabeth Fabre, Etienne Giroux-Leprieur, Jacques Cadranel, Marie Wislez

**Affiliations:** 1Oncology Thoracic Unit, Pulmonology Department, AP-HP, Hôpital Cochin, F-75014 Paris, France; samy.chelabi@aphp.fr (S.C.); xavier.mignard@aphp.fr (X.M.); 2Biochemistry Department, AP-HP, Hôpital Européen Georges Pompidou, F-75015 Paris, France; karen.leroy@aphp.fr; 3Team Inflammation, Complement, and Cancer, Centre de Recherche des Cordeliers, Université de Paris, Sorbonne Université, INSERM, F-75006 Paris, France; 4Department of Pulmonology, Centre Hospitalier Intercommunal de Créteil, F-94000 Créteil, France; Isabelle.monnet@chicreteil.fr; 5Department of Thoracic Oncology, AP-HP, Hôpital Bichat-Claude Bernard, F-75018 Paris, France; solenn.brosseau@aphp.fr; 6Department of Genetic, AP-HP, Hôpital Bichat-Claude Bernard, F-75018 Paris, France; nathalie.theou-anton@aphp.fr; 7Department of Medical Oncology, Institut Curie, F-92210 Saint-Cloud, France; marieange.massiani@curie.fr; 8Department of Pulmonology, Hôpital Foch, F-92150 Suresnes, France; s.friard@hopital-foch.org; 9Department of Thoracic and Medical Oncology, AP-HP, Hôpital Avicenne, F-93000 Bobigny, France; boris.duchemann@aphp.fr; 10Department of medical Oncology, APHP, Hôpital Européen Georges Pompidou, F-75015 Paris, France; elizabeth.fabre@aphp.fr; 11INSERM U970, Université Paris Descartes, F-75015 Paris, France; 12Department of Respiratory Diseases and Thoracic Oncology, APHP, Hôpital Ambroise Paré, F-92100 Boulogne-Billancourt, France; etienne.giroux-leprieur@aphp.fr; 13Team EA4340, BECCOH, Université Paris-Saclay, F-92100 Boulogne-Billancourt, France; 14Pulmonology and Thoracic Oncology Department, AP-HP, Hôpital Tenon, F-75020 Paris, France; jacques.cadranel@aphp.fr; 15Theranoscan GRC 04, Sorbonne Université, F-75970 Paris, France

**Keywords:** NSCLC, EGFR, exon 20 insertion, TKI, chemotherapy, survival

## Abstract

**Simple Summary:**

EGFR exon 20 insertions are rare genetic alterations in non-small-cell lung cancers (NSCLCs) that are usually unresponsive to approved EGFR tyrosine kinase inhibitors (TKIs), but data is limited about this population. We aim to describe clinical features, survival, and response to chemotherapy and TKIs in this patient population.

**Abstract:**

EGFR exon 20 insertions are rare genetic alterations in non-small-cell lung cancers (NSCLCs) that are usually unresponsive to approved EGFR tyrosine kinase inhibitors (TKIs). In this paper, we describe the clinical characteristics, efficacy of EFGR TKIs and chemotherapy, and resulting survival in this population. We retrospectively collected patients with EGFR exon 20 insertions (Exon20ins) from 11 French genetic platforms and paired them (1:2 ratio) with classic Exon 19/21 EGFR mutation patients (controls). Between 2012 and 2017, 35 Exon20ins patients were included. These patients were younger at diagnosis than the controls. All Exon20ins patients who were treated with first-line EGFR TKIs (*n* = 6) showed progressive disease as the best tumor response. There was no significant difference in the tumor response or the disease control rate with first-line platinum-based chemotherapy between the two groups. A trend towards shorter overall survival was observed in Exon20ins vs. controls (17 months (14—not reach(NR) 95% confidence interval(CI) vs. 29 months (17–NR 95%CI), *p* = 0.09), respectively. A significant heterogeneity in amino acid insertion in EGFR exon 20 was observed. EGFR exon 20 insertions are heterogeneous molecular alterations in NSCLC that are resistant to classic EGFR TKIs, which contraindicates their use as a first-line treatment.

## 1. Introduction

Epidermal growth factor receptor (EGFR) mutations, mainly exon 19 deletions and L858R exon 21 mutations, are present in about 12% of lung adenocarcinomas tested in France [1]. Besides these canonical mutations, in-frame insertion mutations in EGFR exon 20 (Exon20ins) are the third-most frequent subtype of EGFR mutation, accounting for up to 4% of EGFR mutations [2]. Exon20ins represent a heterogeneous group of insertions and/or duplications affecting the C-helix domain and the following loop, usually between amino acids 762 and 775. These mutations do not affect the ATP-binding pocket without modifying the affinity of early-generation tyrosine kinase inhibitors (TKIs), compared to wild-type EGFR receptors. Moreover, these insertions/duplications decrease the non-covalent binding of these TKIs through steric hindrance [3]. Exon20ins are known to be generally unresponsive to standard TKIs (erlotinib, gefitinib, afatinib), and their heterogeneity has slowed down the characterization of each alteration and their sensitivity to those TKIs. Recent progress has permitted a comprehensive approach to these molecular alterations and the development of new targeted treatments in both early and advanced phases. A new TKI (mobocertinib) and a specific antibody (amivantamab) recently received FDA breakthrough therapy designations for previously treated Exon20ins NSCLC patients. On-going phase III trials (EXCLAIM-2, NCT04129502 for mobocertinib; and PAPILLON, NCT04538664 for amivantamab) aim to compare these innovative therapies against usual platinum-based chemotherapy. In this period of rapid progress, we aim to improve the characterization of this rare population.

The purpose of our study was to describe EGFR Exon20ins NSCLC patients and evaluate their response to first-line historical ECGF TKIs or chemotherapy.

## 2. Results

### 2.1. Clinical and Molecular Features at Baseline

Between 2012 and 2017, 35 patients with metastatic EGFR Exon20ins NSCLC were included and paired with 76 patients with classic EGFR exon 19/21 deletion/mutation metastatic NSCLC (control group).

Clinical features at baseline are summarized in Table 1.

Exon20ins patients were younger than patients in the control group at diagnosis (mean age 63.8 vs. 69.6 years old, respectively; *p* = 0.02). There was no significant difference in sex, smoking status (active/former smokers vs. non-smokers) or clinical pattern (number of metastatic sites, presence of brain metastases, pleural or pericardial effusion) between the two groups.

EGFR exon 20 insertions are heterogeneous at the molecular level. We found 19 different mutations in the 35 patients harboring EGFR exon 20 insertions, with p.Ala767_Val769dup (*n* = 6) and p.Ser768_Asp770dup (*n* = 6) being the most frequent. No patient harbored any insertions before alanine in position 767, usually considered proximal insertions. Fifty-six patients in the control group had an EGFR exon 19 deletion, and twenty patients had an EGFR exon 21 L858R mutation.

Baseline molecular features are summarized in Table 2. Figure 1 shows comparative data between the CBio Cancer Genomics Portal data and data from the present study. The CBio Cancer Genomics Portal is an open-access database providing access to the genomics data of thousands of tumors from hundreds of cancer studies [4,5]. For NSCLC cancer, the CBio Portal aggregates 20 studies, gathering more than 6000 samples.

### 2.2. Treatment

Treatment characteristics are summarized in Table 3 and Table 4.

Exon20ins patients received chemotherapy significantly more frequently (94% vs. 37%, respectively, *p* < 0.01), and received more lines of chemotherapy (2.2 vs. 1.3 lines, respectively, *p* < 0.01), than the control group. Conversely, the control group received EGFR TKIs significantly more frequently (97% vs. 60%, respectively, *p* < 0.01) and received more TKI lines (1.6 vs. 1.1 lines, respectively, *p* < 0.01).

The first-line treatment was significantly different. Chemotherapy was the first-line treatment in 27/35 (77%) Exon20ins patients, vs. 8/76 (11%) patients in the control group (*p* < 0.01). Conversely, 6/35 (17%) Exon20ins patients received EGFR TKIs as the first-line treatment vs. 63/76 (82%) in the control group (*p* < 0.01). Two patients in the exon 20 insertion group and five patients in the control group received no treatment.

### 2.3. Response to EGFR TKIs

Tumor responses to EGFR TKIs are summarized in Table 5.

The response to EGFR TKIs differed significantly between the two groups, with a 0% objective response rate (ORR) in the Exon20ins group vs. 68% in the control group (*p* < 0.01). Disease control (DC) with first-line EGFR TKIs was 0% in the Exon20ins group vs. 84% in the control group with a complete response (CR) in 3%, a partial response (PR) in 65%, and stable disease (SD) in 16% (*p* < 0.01).

Moreover, among Exon20ins patients receiving first- or second-generation EGFR TKIs at any line (*n* = 19 with evaluable response), none showed ORR.

### 2.4. Response to Platinum-Based Chemotherapy

Tumor response after platinum-based chemotherapy is summarized in Table 5.

There was no significant difference in tumor response with the first-line platinum-based chemotherapy between the two groups. The DC was 82% in the Exon20ins group vs. 64% in the control group (*p* = 0.3).

### 2.5. Overall Survival

There was a trend towards shorter overall survival (OS) in the Exon20ins group compared to the control group (median survival 17 (14–NR, 95%CI) months vs. 29 (27–NR, 95%CI) months, respectively (*p* = 0.09)), as shown in Figure 2.

### 2.6. Outlier Long Survivors in Exon 20 Insertion Group

Four patients in the Exon20ins group had a markedly longer OS ≥ 24 months. The types of insertions and the treatment these patients received are presented in Table 6. All other patients (*n* = 31) had OS ≤ 18 months, with a median OS of 17 months. Among the long survivors, patient #1 (Asn771_Pro772insCysAlaTyr) received afatinib for 9 months with stable disease, patients #2 and #3 were heavily treated with three and five lines of chemotherapy/immunotherapy, respectively, followed by one course of EGFR TKI. The last patient (#4) had SD for 18 months under first-line chemotherapy with carboplatin-pemetrexed/pemetrexed maintenance. None of these mutations are known to be associated with a better prognosis.

### 2.7. Progression-Free Survival

All patients with first-line TKIs in the Exon20ins group (*n* = 6) showed PFS < 3 months, and progressive disease was the best response (Table 5). For first-line TKIs, PFS was 2 (2–2, 95% CI) vs. 6.5 (4–12, 95%CI) months for first-line chemotherapy (*p* < 0.001) (Figure 3 and Appendix A). Conversely, all patients with first-line PFS ≥ 6 months (*n* = 13) received platinum-based chemotherapy.

In the control group, first-line TKIs PFS was 12 (8–16, 95%CI) vs. 2.5 (2–2.5, 95% CI) months for first-line chemotherapy (*p* < 0.001) (Figure 3 and Appendix A).

Among Exon20ins patients, PFS was not different between patients with chemotherapy + bevacizumab vs. patients with chemotherapy alone (Appendix A).

## 3. Discussion

The aim of our study was to describe Exon20ins NSCLC patients and their response to historical EGFR TKIs and platinum-based chemotherapy. Between 2012 and 2017, thirty-five Exon20ins patients were included in our study. These patients were quite comparable with classic EGFR mutation patients except they had younger ages at diagnosis.

Overall survival appeared to be longer in the classic exon 19/21 group than in the Exon20ins group with a 12 month gap in median OS (29 months vs. 17 months). However, this was not statistically significant (*p* = 0.09), probably due to the small number of patients and the few outlier long survivors in the Exon20ins group. All four Exon20ins long survivors harbored a different insertion, forbidding any molecular explanation for this outcome.

The 17 month OS reported here is consistent with previous observations [6,7]. Although they are controversial, some retrospective studies and post-hoc analyses in small groups of patients, such as the study by Naidoo et al. (*n* = 46), found no difference in OS between patients with exon 19/21 mutations and Exon20ins (31 months vs. 26 months, respectively, *p* = 0.53) [8].

The PFS was significantly lower with first-line TKIs than with chemotherapy in the Exon20ins group. None of our Exon20ins patients who received first- or second-generation EGFR TKIs at any line (*n* = 19 with evaluable response) showed ORR. At first line, all of those patients (*n* = 6) showed PD as the best response at first assessment. Among the four Exon20ins patients experiencing an OS ≥ 24 months, one (p.Asn771_Pro772insCysAlaTyr) showed an unusual nine month SD with afatinib. In our study, eight patients received afatinib (*n* = 3 at first-line treatment and *n* = 5 at following lines). Among these patients, five showed PD as the best response (including the three first-line patients), two patients showed stable disease and one had a non-evaluable response). These results are consistent with Yang et al.’s combined post-hoc analysis of patients treated with afatinib in LUX-Lung 2, LUX-Lung 3 and LUX-Lung 6 (*n* = 23 patients, PFS = 2.7 months, and disease control rate = 15%), and other retrospective studies [7,8,9,10,11,12]. These data support the well-known data indicating that early-generation TKIs are not effective in EGFR Exon20ins NSCLC.

None of the Exon20ins patients in our cohort received osimertinib due to the period of study. In vitro and in vivo models showed promising antitumor activity of osimertinib in Exon20ins patients [13]. Pharmacological studies suggest that higher dosing may be required to achieve clinical efficacy in these populations [14]. A recent prospective phase II trial, ECOG-ACRIN EA5162, evaluated a 160 mg daily dosing of osimertinib in previously treated Exon20ins patients. In this trial, Piotrowska et al. reported a DCR of 82% and 9.6 months median PFS [15].

Among Exon20ins patients, PFS appeared to be longer in those treated with chemotherapy + bevacizumab vs. chemotherapy alone (Appendix A). This is consistent with a very large retrospective Chinese study [16].

In our study, PFS in the control group who received first-line chemotherapy (PFS = 2.5 months) seemed lower than in large phase III trials that showed PFS ranging from 4.6 to 5.8 months [17,18,19,20], probably due to a smaller group (*n* = 6) and unselected population. For these reasons, we can allow neither further interpretation on PFS, nor a frontal comparison between groups on first-line efficacy based on our study.

New TKIs specially designed to target EGFR exon 20 insertion mutations are under investigation and are already considered serious treatment options for Exon20ins NSCLC. Although poziotinib showed disappointing results (low ORR and limiting toxicities) [21,22], mobocertinib recently obtained breakthrough therapy designation from the U.S. FDA after a phase I/IItrial and expansion cohort (PFS = 7.3 months and ORR = 26%) [23]. Phase II EXCLAIM confirmed PFS = 7.3 months, and quite similarly, ORR = 23% [24]. The EXCLAIM-2 phase III trial, evaluating mobocertinib vs. platinum-based chemotherapy in treatment-naïve patients, is ongoing (NCT04129502). Other TKIs, such as CLN-081 and DZD9008, recently showed interesting ORR (40% and 48%, respectively) in pre-treated Exon20ins patients with acceptable safety profiles [25,26].

Different targeted strategies have been developed. The bi-specific anti-EGFR and anti-MET antibody amivantamab showed exciting results (PFS = 8.3 months and ORR = 40%) with an apparently acceptable safety profile) [27]. Amivantamab received FDA breakthrough therapy designation in March of 2020 and became the first treatment to be granted FDA approval in May of 2021 for patients with locally advanced or metastatic NSCLC with EGFR exon 20 insertion mutations after platinum-based chemotherapy.

In present study, all insertions occurred in the loop following the C-helix, after amino acids at position 767. None of the patients harbored insertions before the 764 amino acid position, which could have conferred sensitivity to approved EGFR TKIs for canonical exon 19 and 21 mutations, in particular the p.Ala763_Tyr764insPheGlnGluAla insertion (A736-Y764insFQEA) [7,18]. Among 35 patients, we collected 18 different Exon20ins variants. We found p.Ala767_Val769dup (*n* = 6) and p.Ser768_Asp770dup (*n* = 6) as the more frequent insertion mutations. This data is consistent with the CBio Cancer Genomics Portal database, as presented in Figure 1.

Given the heterogeneity of Exon20ins identified in our study and in previous reports, the efficacy of targeted treatments may vary across different variants even if early trial data did not distinguish a clear correlation between Exon20ins variants and the efficacy of mobocertinib or amivantamab. Further studies of the behavior variability of new targeted therapies against different variants are needed in order to provide better comprehension of acquired resistances after Exon20ins targeted therapies.

New TKIs and specific antibodies are already effective therapies for exon 20 insertion NSCLC patients. The large phase III trials EXCLAIM-2 and PAPILLON will respectively clarify the position of mobocertinib and amivantamab vs. standard chemotherapy. The relevance of immune-checkpoint inhibition alone or in combination with platinum-based chemotherapy is largely unknown, and future trials may enrich treatment options in this population.

## 4. Materials and Methods

### 4.1. Study Design

We retrospectively collected all consecutive patients with EGFR Exon20ins metastatic NSCLC from 11 genetic platforms in France between 2012 and 2017. When available, we paired each Exon20ins patient with 2 metastatic EGFR classical L858R exon 21 mutation or exon 19 deletion NSCLC patients (controls). To ensure consistency of care, we chose controls treated in the same care facility and diagnosed in the same 3-month period. If more than 2 controls met these criteria, we randomly chose 2 of them. Exclusion criteria were localized or locally advanced NSCLC, patients in whom EGFR status was not obtained at a primary biopsy (i.e., re-biopsy at progression after systemic treatment), and the presence of concomitant active neoplasia.

EGFR TKIs or conventional platinum-based chemotherapy as a first- or subsequent-line therapy was at the physician’s discretion. Disease assessment was based on a CT scan every two or three months and was also at the physician’s discretion.

Between 2012 and 2017, EGFR exon 20 insertions were detected from a formalin-fixed paraffin-embedded biopsy by PCR with fluorescent primers and fragment analysis (Sanger). Next-generation sequencing (NGS) was used only at the end of our period of study (2017).

### 4.2. Efficacy Analysis

The main objective of this study was to evaluate the efficacy of first-line EGFR TKIs and conventional chemotherapy in patients with metastatic EGFR Exon20ins NSCLC, defined by the objective response rate (ORR). Tumor response was assessed according to the Response Evaluation Criteria in Solid Tumors Version 1.1 (RECIST 1.1). Tumor responses included a complete response (CR), partial response (PR), stable disease (SD) and progressive disease (PD). The disease control rate was defined as the addition of complete responses, partial responses and stable disease (CR + PR + SD). The objective response rate (ORR) was defined as the addition of complete responses and partial responses (CR + PR).

The secondary aims of our study were to determine overall survival (OS) and progression-free survival (PFS).

### 4.3. Statistical Analysis

OS and PFS were estimated using the Kaplan–Meier method. PFS was calculated from the date treatment started until death or disease progression. OS was calculated from the date of the molecular diagnosis of the EGFR mutation until death or the final follow-up (in patients that were lost to follow-up) or the date of censorship (in living patients). The date of censorship was 1 April 2018. OS and PFS were compared using the log-rank test between the two groups. All statistical analyses were performed using R-studio software version 1.1.419 (The R Foundation for Statistical Computing, Vienna, Austria).

## 5. Conclusions

EGFR Exon20ins are heterogeneous genetic alterations conferring resistance to first- and second-generation EGFR TKIs in NSCLC. Patients showed similar clinical characteristics to patients with classic EGFR exon 19/21 mutations but with a tendency to poorer overall prognosis. Platinum-based chemotherapy should be considered as the first-line treatment in these patients until the results of an ongoing trial in exon 20 targeted TKIs become available.

## Figures and Tables

**Figure 1 cancers-13-05132-f001:**
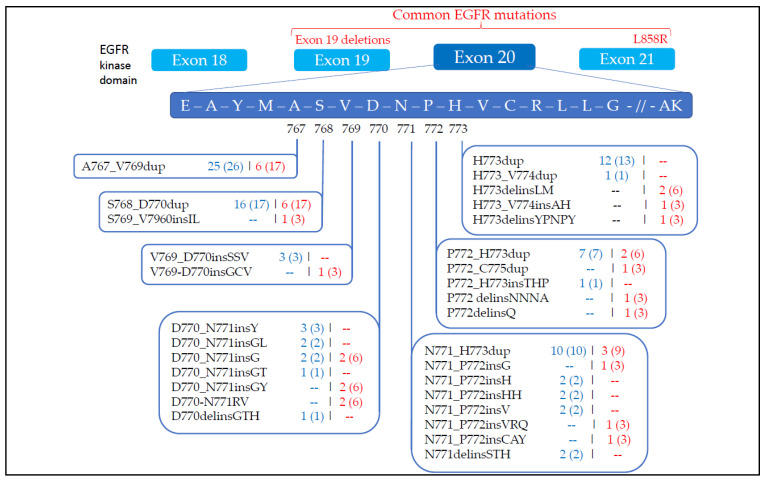
Comparative prevalence of EGFR exon 20 insertions in NSCLC between the CBio Cancer Genomics Portal database and the present study. Detailed insertions from amino acid positions 767 to 773 in EGFR exon 20: in blue, exon 20 insertions from CBio Cancer Genomics Portal, *n*(%); in red, detailed exon 20 insertions from our study, *n*(%).

**Figure 2 cancers-13-05132-f002:**
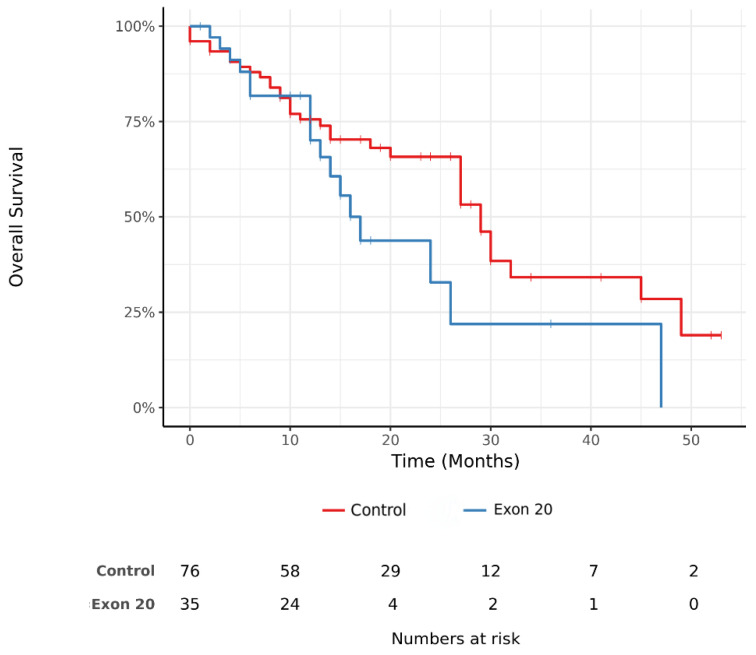
Kaplan–Meier estimation of overall survival (OS) measured from the inclusion of patients in the Exon20ins and control groups.

**Figure 3 cancers-13-05132-f003:**
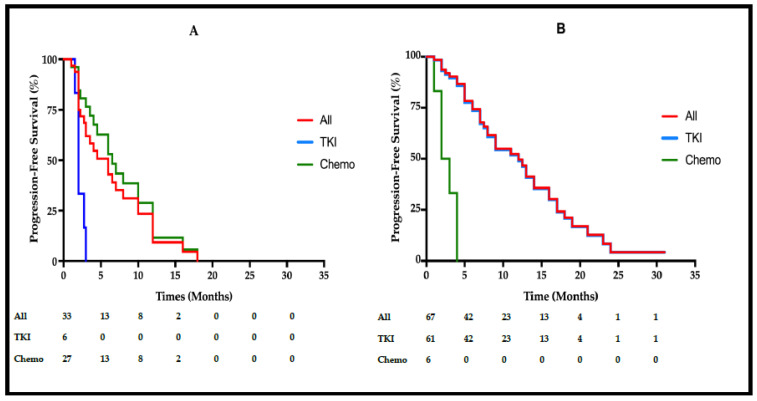
Kaplan–Meier estimation of progression-free survival (PFS) measured from inclusion in patients with exon 20 insertion (**A**) and control patients (**B**). PFS was determined for all patients (red), first-line TKIs patients (blue), and first-line chemotherapy patients (green) in each group.

**Table 1 cancers-13-05132-t001:** Clinical features of patients from the Exon20ins group and the control group at inclusion.

Variables		Exon20ins ^1^ Group *n* = 35	Control Group *n* = 76	*p*-Value
		*n* (%)		
Age (y)		63.8	69.9	**0.02** *
Gender	F/M	19/16 (54/46)	54/22 (71/29)	0.08 ^#^
ECOG ^2^				0.6 ^#^
	0–1	29 (83)	56 (74)	
	≥2	5 (14)	13 (17)	
	Unknown	1 (3)	7 (9)	
Smoking status				0.6 ^#^
	Active or former	14 (40)	27 (36)	
	Never	20 (57)	47 (62)	
	Unknown	1 (3)	2 (2)	
Mean tobacco consumption	Pack-years	21	27	0.49 *
Pleural effusion		16 (46)	39 (51)	0.5 ^#^
Pericardial effusion		0 (0)	6 (8)	0.17 ^§^
TNM classification (7th edition)				
T indicator				0.6 ^§^
	Tx	4 (11)	15 (20)	
	T1	1 (3)	7 (9)	
	T2	12 (34)	19 (25)	
	T3	7 (20)	13 (17)	
	T4	11 (32)	22 (29)	
N indicator				0.2 ^§^
	Nx	5 (14)	11 (15)	
	N0	2 (6)	17 (22)	
	N1	3 (9)	6 (8)	
	N2	12 (34)	23 (30)	
	N3	13 (37)	19 (25)	
M indicator				0.74 ^#^
	M1a	11 (31)	19 (25)	
	M1b	8 (23)	17 (22)	
	M1c	16 (46)	40 (53)	
Numbers of metastatic sites	Mean number	1.89	2.26	0.16 *
Brain metastasis	Brain	10 (29)	25 (33)	0.5 ^#^

^1^ EGFR exon 20 insertion; ^2^ Eastern Cooperative Oncology Group. *p*-Value based on * Student’s *t*-test, ^#^ chi² test and ^§^ Fisher test.

**Table 2 cancers-13-05132-t002:** Amino acid positions of insertions in exon 20 of the EGFR gene.

Amino Acid in EGFR Exon 20	EGFR Exon 20 Insertions (*n* = 35)	Type of Insertion	*n*
A767	17% (*n* = 6)	p.Ala767_Val769dup	*n* = 6
S768	20% (*n* = 7)	p.Ser768_Asp770dup p.Ser768_Val769delinsIleLeu	*n* = 6 *n* = 1
V769	3% (*n* = 1)	p.Val769_Asp770insGlyCysVal	*n* = 1
D770	17% (*n* = 6)	p.Asp770_Asn771insArgVal p.Asp770_Asn771insGly p.Asp770delinsGlyTyr	*n* = 2 *n* = 2 *n* = 2
N771	17% (*n* = 6)	p.Asn771_Pro772insGly p.Asn771_Pro772insValArgGln p.Asn771_Pro772insCysAlaTyr p.Asn771_His773dup	*n* = 1 *n* = 1 *n* = 1 *n* = 3
P772	14% (*n* = 5)	p.Pro772_His773delinsGln p.Pro772delinsAsnAsnAsnAla p.Pro772_His773dup p.Pro772_Cys775dup	*n* = 1 *n* = 1 *n* = 2 *n* = 1
H773	11% (*n* = 4)	p.His773_Val774insAlaHis p.His773_Val774delinsLeuMet p.His773delinsTyrProAsnProTyr	*n* = 1 *n* = 2 *n* = 1

**Table 3 cancers-13-05132-t003:** Lines of treatment during follow-up in the exon 20 insertion group and the control group.

Treatment Lines *n* (%)	Exon20ins ^1^ Group *n* = 35 (%)	Control Group *n* = 76 (%)	*p*-Value
	*n* (%)		
**TKIs ^2^**			
TKIs administration	21 (60)	74 (97)	<0.01 ^#^
1 TKI line	19 (54)	41 (54)	
2 TKI lines	1 (3)	21 (28)	
≥3 TKI lines	1 (3)	10 (13)	
**Chemotherapy**			
Chemo administration	33 (94)	28 (37)	<0.01 ^#^
1 Chemo line	13 (37)	20 (26)	
2 Chemo lines	7 (20)	5 (7)	
≥3 Chemo lines	12 (34)	1 (2)	

^1^ EGFR exon 20 insertion; ^2^ tyrosine kinase inhibitors. *p*-Value based on ^#^ chi² test.

**Table 4 cancers-13-05132-t004:** First-line treatment in Exon20ins group and in control group.

Type of First-Line Treatment	Exon20ins ^1^ Group *n* = 35	Control Group *n* = 76
	*n* (%)	
TKIs ^2^		
Erlotinib	3	19
Gefitinib	0	35
Afatinib	3	9
Chemotherapy		
Platin–pemetrexed	16	7
Platin–pemetrexed–bevacizumab	5	1
Platin–paclitaxel	1	0
Platin–paclitaxel–bevacizumab	3	0
Platin–vinorelbine	1	0
Platin–pemetrexed–nivolumab	1	0
Other		
No treatment	2	5

^1^ EGFR exon 20 insertion; ^2^ tyrosine kinase inhibitors.

**Table 5 cancers-13-05132-t005:** Tumor responses with first-line treatment in the Exon20ins and control groups.

Variation	First-Line TKIs ^1^		*p*-Value	First-Line Chemotherapy		*p*-Value
	Exon20ins ^2^ *n* = 6 (%)	Control *n* = 63 (%)		Exon20ins ^2^ *n* = 27 (%)	Control *n* = 8 (%)	
**Complete Response (CR)**	0	2		0	0	
**Partial Response (PR)**	0	41		11	1	
**Objective response rate (ORR) = CR + PR**	**0%**	**68%**	**<0.01 ^§^**	**41%**	**12%**	**0.2 ^§^**
**Stable disease (SD)**	0	10		11	4	
**Disease Control (DC) = CR + PR + SD**	**0%**	**84%**	**<0.01 ^§^**	**82%**	**64%**	**0.3 ^§^**
**Progressive Disease**	6 **100%**	3 **5%**		3 **11%**	2 **24%**	
**Non Assessable (NA)**	0 **0%**	7 **11%**		2 **7%**	1 **12%**	

^1^ Tyrosine kinase inhibitors; ^2^ EGFR exon 20 insertion; **^§^**
*p*-Value based on Fisher test.

**Table 6 cancers-13-05132-t006:** Detailed clinical, molecular, and treatment features of outsider long survivors in Exon20ins group.

Patients	Sex/Age/ECOG	Brain Metastasis	TNM (7th ed.)	Type of Exon 20 Insertions	Treatments	OS (Months)
**1**	F/68/0	Yes	cT2N2M 1c	p.Asn771_Pro772insCysAlaTyr	Cisplatin–pemetrexed–bevacizumab (M) ^1^ Afatinib	24
**2**	F/53/0	Yes	cT3N1M 1b	p.Asn771_His773dup	Cisplatin–navelbin Paclitaxel–bevacizumab Carboplatin–pemetrexed Erlotinib	26
**3**	F/65/1	Yes	cT2N2M 1c	p.Ala767_Val769dup	Carboplatin–paclitaxel–bevacizumab (M) ^1^ Pemetrexed Docetaxel Carboplatin–pemetrexed (M) ^1^ Nivolumab Erlotinib	47
**4**	M/72/1	No	cT3N3M1b	p.Asn771_Pro772insGly	Carboplatin–pemetrexed (M) ^1^ Nivolumab Afatinib	36

^1^ (M): Maintenance of previous drug.

## Data Availability

The datasets analyzed during the current study are available from the corresponding author upon reasonable request.

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
