# Peer review of "EGFR Exon 20 Insertion in Metastatic Non-Small-Cell Lung Cancer: Survival and Clinical Efficacy of EGFR Tyrosine-Kinase Inhibitor and Chemotherapy"

_cancers, 2021, doi:10.3390/cancers13205132_

Round 1

Reviewer 1 Report

In the current report Chelabi and colleagues report on clinical outcomes of 35 patients with EGFR exon 20 insertion mutations and 76 patients with classical EGFR mutations. The authors preform a retrospective analysis of eleven French platforms to study the outcomes of patients with EGFR exon 20 insertion mutations or classical EGFR mutations after treatment with first line chemotherapy or EGFR TKI. Overall the manuscript is well written, easy to understand, and provides important information on a subpopulation of underserved patients with difficult to treat EGFR mutations. However, the limited number of patients, particularly patients with classical EGFR mutations treated with first line chemotherapy, limits the interpretation of the results.  

Major Concerns:

  1. The main conclusion of the paper is that patients with EGFR exon 20 insertion mutations have a shorter overall survival than patients with classical EGFR mutations; however, the difference between groups was not statistically significant. While an intuitive finding, as others have previous reported on the low response rates to targeted therapy, the results are an important baseline for the field. Therefore, increasing the statistical power of the analysis is important.

  1. The authors report on the low frequency of EGFR exon 20 insertions with only 35 patients detected over five years across eleven databases. This seems very low. What was the method of detection for determining EGFR exon 20 mutant patients in this cohort? What was the total number of patients screened during this period? This should be included in the methods.

Minor Concerns:

  1. PFS of both cohorts should be compared directly in figure 2 since OS does not have statistical power to determine differences between cohorts.

  1. The ORR and PFS of patients with classical EGFR mutations treated with chemotherapy seems low compared to other studies in the literature. Increasing the number of patients in this cohort could further strengthen the finding that there is no difference between patients classical and exon 20 mutations.

  1. When describing the exon 20 insertion mutations, at times the actual inserted amino acids are missing (line 127 and 167). The amino acids inserted should be included along with the location of the mutation. Similarly, for duplications, while implied, amino acids duplicated should be included with the name of the mutation.

  1. Table 6 is difficult to read. Additional space should be included between patients so that the treatments each patient received is distinguishable. Currently, it is difficult to determine which treatment corresponds to which patient, particularly the first and last patient.

  1. In table 6, two of the four highlighted patients with prolonged responses received bevacizumab. Were these the only two patients in the 35 to receive a VEGF inhibitor? Did treatment with a VEGF inhibitor prolong the PFS in the greater population? Current clinical studies and preclinical studies are unclear on the benefit of VEGF inhibition in the front-line setting, so this analysis could interesting if the data is available.

  1. In the discussion, clinical outcomes for mobocertinib, poziotinib, and amivantamab should be updated to the latest results.
    1. Mobocertinib updated data showed a 23% confirmed ORR (https://www.jto.org/article/S1556-0864(21)00325-7/fulltext).
    2. Poziotinib has updated the dosing regimen and presented preliminary results at AACR 2021 (Le X, Shum E, Suga J, et al. Presented at: AACR Annual Meeting 2021; April 10-15, 2021; virtual. Abstract CT169.).
    3. Amivantamab included an update at on AEs at WCLC https://www.jto.org/article/S1556-0864(21)00326-9/fulltext.

Reviewer 2 Report

Overall, a good manuscript.  Certainly, exon 20 insertions are a topic of interest, given the emergence of experimental therapies that have shown activity in this population.  Further, there is limited available real world experience with this patient population.

Some concerns:

1.) Emphasis is placed on the 6 patients that got EGFR TKI in the exon 20 group first line, and all progressed.  I am more interested in the 20+ patients that got EGFR inhibitors at any line.  6 patients is quite few to draw conclusions.

2.) Afatinib is the drug that may have the most data in this population, I am interested to know how afatinib did in this patient population; 2 of the patients with extended survival appeared to recieve afatinib.  More details are needed on response rate to afatinib.

3. Figure 2, should have the same X-axis; right now the values are different which can be deceptive.

4.) Some of discussion is just re-hashing of results section.  

5.) Some of discussion is a bit over-stated; the results of 6 patients that got first line TKI does not "confirm" much ("these data clearly confirm...).  However, the data does support these conclusions...

6) The following sentences need clarification, eg are you referring to patients with exon 20 insertions (presumably)?  It is not clear as written.

None of the patients received osimertinib in our cohort. Fang and al. reported a 100% disease control rate (DCR) and PFS=6.2 months 178 in six patients treated with osimertinib [11]. Piotrowska and al. reported consistent data 179 with 160mg daily of osimertinib with a DCR of 82% and 9.6 months median PFS [12]. 

7.) Similarly, inappropriate to draw conclusions on efficacy of first line chemo in patients with egfr 19/21 mutations with only 6 patients.  Level of evidence on this issue is not of sufficient robustness for publication in this journal, as the author points out.  Would remove this from discussion.

8.) The following sentence is not well written:

"Initially promising results with poziotinib have been somewhat disappointing" 

9.) Not clear how the exon 20 patients were paired with patients with classical egfr mutations.  Were the patients with classical egfr mutations randomly chosen?

10.) In efficacy analysis of methods section, authors describe "secondary goals" and I am not familiar with that term.

In summary, improvements are needed.  But given the paucity of available data on patients with exon 20 insertions treated in a real world setting, I do believe this is an important manuscript with value to the readership of the Journal.

Round 2

Reviewer 1 Report

In the revised manuscript Chelabi and colleagues report on clinical outcomes of 35 patients with EGFR exon 20 insertion mutations and 76 patients with classical EGFR mutations. The authors preform a retrospective analysis of eleven French platforms to study the outcomes of patients with EGFR exon 20 insertion mutations or classical EGFR mutations after treatment with first line chemotherapy or EGFR TKI. The small number of patients included limits the interpretation of the results. While the authors did not address this major concern directly, they do adjust the language of the conclusions to reflect this caveat of the study. Other concerns were largely addressed. Only minor issues below.

Minor concerns:

  1. Please include the different methods used to detect EGFR mutations in the methods section (as stated in response to reviewers response 2, point 4).
  2. Accelerated FDA-approval of amivantamab should be noted.
  3. CLN-081 and DZD9008 preliminary data could be mentioned in the conclusion with other inhibitors.
  4. Discussion of osimertinib for patients with exon 20 insertions should be updated to recent published reports: (Yan et al 2021 Lung Cancer, Veggel et al 2020 Lung Cancer).
  5. With the tracked changes, there are a few typos in the text. 
    1. awkward phrasing line 181-182
    2. two periods line 210
    3. pleural on "months" line 276

Reviewer 2 Report

Still a good paper with important implications.  However, continues to have some stylistic weaknesses. 

Authors seem to place some results in the discussion section, eg Supplementary Figure 2 is only discussed in Discussion section.  Also, the following results are only in discussion section, and would seem to be more appropriate for Results section:

"None of our Exon20ins patients receiving 1st or 2nd gener-
ation EGFR TKI at any line (n=19 with evaluable response) showed ORR (data not shown)."

" molecular analysis revealed that all the insertions in our study occurred in 
the loop following the C-helix, after the 767 amino-acid position, which are classically described as preferential sites of insertion."

Author Response

Please see the attchement
